# Adherence to Anti-Osteoporotic Treatment and Clinical Implications after Hip Fracture: A Systematic Review

**DOI:** 10.3390/jpm11050341

**Published:** 2021-04-24

**Authors:** Ramona Dobre, Dan Alexandru Niculescu, Răzvan-Cosmin Petca, Răzvan-Ionuț Popescu, Aida Petca, Cătălina Poiană

**Affiliations:** 1“Carol Davila”, University of Medicine and Pharmacy, 050474 Bucharest, Romania; dan.niculescu@umfcd.ro (D.A.N.); razvan.petca@umfcd.ro (R.-C.P.); razvan-ionut.popescu@drd.umfcd.ro (R.-I.P.); aida.petca@umfcd.ro (A.P.); catalina.poiana@umfcd.ro (C.P.); 2Department of Endocrinology, National Institute of Endocrinology CI Parhon, 011853 Bucharest, Romania; 3Department of Urology, “Prof. Dr. Th. Burghele” Clinical Hospital, 050659 Bucharest, Romania; 4Department of Obstetrics and Gynecology, Elias University Hospital, 011461 Bucharest, Romania

**Keywords:** anti-osteoporotic treatment, hip fracture, mortality, re-fracture risk

## Abstract

The role of anti-osteoporotic treatment as part of the secondary prevention after hip fracture in terms of mortality and re-fracture risk has been studied, and the results are promising. Decreased treatment adherence and compliance is a problem that needs to be addressed by healthcare professionals. A systematic review of the literature was performed using the PubMed database with terms that included hip fracture, mortality, second fracture, and specific anti-osteoporotic treatment. We included 28 articles, 21 regarding mortality and 20 re-fracture rates in hip fracture patients. All studies showed lower mortality after hip fracture associated with anti-osteoporotic treatment, mostly bisphosphonate agents. The re-fracture risk is still debatable, since conflicting data were found. Although most of the studies showed notable effects on mortality and re-fracture rates associated with anti-osteoporotic treatment, we still need more data to validate the actual results.

## 1. Introduction

Hip fracture remains the most important clinical manifestation of osteoporosis, associated with high mortality and morbidity with significant socioeconomic consequences.

Relevant steps were made in the last decade to increase the diagnostic rate of osteoporosis. This pathology’s silent character means that the fragility fracture, especially since hip fracture has the noisiest clinical picture, is the first presentation of the disease in a substantial number of cases, regardless of age. At the same time, most of the international guidelines for osteoporosis diagnosis recommend screening in women after 65 years of age and in men after 70 years, leaving the population between 50 and 65 years at risk of directly developing the clinical signs of osteoporosis, namely fragility fractures [1].

It is recognized that a fragility fracture is an essential risk factor for a second fracture [2], with a two times higher relative risk of sustaining a new vertebral or hip fracture after a prevalent fracture. This risk is considered to be higher immediately after the index fracture and decreases with time [3]. Studies demonstrated a linear relationship with age [4]. This effect is more pronounced in the elderly population [4].

The most prominent consequence of hip fracture is the associated higher mortality, which has remained virtually unchanged in the last few decades [5]. Almost a third of these patients will die within one year after a hip fracture [6]. Studies that analyzed the trends in mortality rates after hip fracture showed that the risk decreases with time [6]. The critical period for these patients is the first year after the fracture, with the rates remaining higher even in the following years compared with the general population. Although it is widely accepted that the excess mortality after hip fracture is linked to the pre-fracture comorbid status of the patient and post-fracture complications, studies that adjusted for these factors showed unexplained excess mortality [7].

The secondary prevention of fractures that incorporates medical treatment to improve bone health is essential in breaking the vicious cycle of fragility fracture in these patients. At the same time, most healthcare systems do not succeed in implementing the necessary measures after the first fracture in order to prevent the second one. Only 9–20% of patients receive anti-osteoporotic treatment [8,9,10].

In 2003, one of the first studies was published that showed a decreased risk of mortality in treated patients with fragility fractures with an odds ratio of 0.25 (CI 95% 0.006–1.12) for one-year and 0.34 (CI 95% 0.17–0.70) for long-term mortality risk compared with non-treated patients [11]. One of the most cited articles regarding clinical fractures and mortality after hip fractures was published in 2007, where the authors demonstrated a 28% reduction in mortality rate in patients receiving zoledronic acid compared to placebo (*p* = 0.01) [12] and, also, lower rates of subsequent fragility fractures in the treatment group compared to placebo.

The present study aims to offer a comprehensive review of the literature regarding the effect of anti-osteoporotic treatment on mortality and also in preventing secondary fragility fractures in patients with hip fractures.

## 2. Materials and Methods

The PRISMA guidelines (preferred reporting items for systematic reviews and meta-analysis, 2015) were followed in the literature review process.

The PubMed, SCOPUS, and clinicaltrials.gov databases were used for a comprehensive literature search, from January 2011 to March 2021. For PubMed and Scopus, the following keywords were used: hip fracture mortality and bisphosphonate (1a), teriparatide (1b), denosumab (1c), and anti-osteoporotic treatment (1d), and second fracture and bisphosphonate (2a), teriparatide (2b), and denosumab (2c) and anti-osteoporotic treatment (2d). For clinicaltrials.gov, we chose a more extensive search using only hip fracture mortality (3a) and second fracture (3b) in studies that started from January 2011. The inclusion criteria were hip fracture patients, with history of specific anti-osteoporosis treatment (bisphosphonates, denosumab, and teriparatide) before or after the index fracture and related outcomes such as mortality and second fragility fracture (hip or other major fracture). Age and sex were not considered inclusion criteria. We also searched relevant cited articles found within articles. In vitro studies, editorials, letters to the editor, case reports, commentaries, and animal studies were excluded. The PRISMA flow diagram is shown in Figure 1.

Descriptions of the studies included in the review, with their main findings and limitations, are shown in Table 1. Data regarding characteristics and limitations of each included study are shown in Appendix A.

## 3. Adherence to Treatment

Anti-osteoporotic treatment plays an important role both in the primary and secondary prevention of fragility fractures. Bisphosphonates (BPs) are used as a first-line therapy and are widely recognized as efficient and safe [41]. BPs prevent fractures by increasing bone mineral density (by suppressing bone turnover rates) and are associated with a 40–70% reduction in vertebral and hip fracture rates [42]. Another anti-resorptive agent used is denosumab, a human monoclonal antibody to RANKL, first approved for use in 2010 [43]. Like the BP treatment, denosumab was also proven to effectively prevent osteoporotic fractures, with a significant reduction in vertebral and non-vertebral fragility fractures [35,44]. In Europe, adherence to denosumab is higher compared to that to the BP treatment [45], which is probably explained by the administration mode—denosumab requires only a subcutaneous dose at six months compared to a weekly/monthly administration of oral BP [46]. The authors showed a 24-month persistence with denosumab of 75.1% to 86% with adherence of 62.9% and 70.1%, with lower durability in patients who had at least one fall in the last 12 months and subjects with more comorbidities [45].

In Romania, a fragility fracture is eligible for reimbursed anti-osteoporotic treatment only when combined with a bone mass density T score of at least −2 standard deviations measured with DXA scan (dual-energy X-ray absorptiometry). The deficient availability of DXA units in our country further decreases the proportion of hip fracture patients that will receive anti-osteoporosis treatment.

In 2017, an Expert Consensus from the European Society of Clinical and Economic Aspects of Osteoporosis and Osteoarthritis (ESCEO) highlighted that only a fifth of patients that sustained a fragility fracture received anti-osteoporotic treatment, with a relevant heterogenicity between regions and countries [47].

The National Hip Fracture Database is the largest hip fracture database in the world, with more than 650,000 hip fractures registered. Due to the development of national clinical standards and the implementation of Orthogeriatric Services, in 2016, there was a reported rate of 60% of patients who received anti-osteoporotic treatment after hospital discharge [47]. Even in hip fracture patients who receive anti-osteoporotic therapy, compliance to treatment is another problem that healthcare professionals need to address. In the UK, the Best Practice Tariff, a paying model for the medical services based on the standard level of care, recommends, aside from the improvements in the treatment care in hip fracture patients, the collection of compliance to treatment data through telephone interrogation 120 days after hospital discharge [48].

In a study that aimed to observe geographic variation in anti-osteoporosis therapy in the UK [49], the authors showed that in 1999, only 5% of hip fracture patients received anti-osteoporosis treatment after the index fracture. Between 1999 and 2004, the percentage increased to 10%, rising spectacularly to 40% between 2005 and 2013. Between 2011 and 2013, a downward trend occurred, with the percentage dropping from 51 to 39% [49].

One study analyzed anti-osteoporotic drug use during four years in Austria [17], covering 98% of the entire population. It showed that only 37.69% of patients with a hip fracture between 2008 and 2010 received anti-osteoporotic treatment, of which 25.9% received BP (18.41% alendronate, 6.49% risedronate, and 5.18% ibandronate). The remaining 62.31% did not receive any anti-osteoporosis treatment one year before and six months after the hip fracture [17]. A higher percentage of women received BP treatment compared to men (30.98% versus 12%). BP was administered before the hip fracture in 20.65% of women compared with 6.11% of men. After the index fracture, 10.34% of women compared to 5.89% of men began treatment with BP, signifying that osteoporosis is underdiagnosed and undertreated in male patients compared to women [17].

Cobden et al. [19] described that 460 out of 562 (82%) patients with hip fracture treated with hemiarthroplasty did not receive anti-osteoporotic treatment and screening in the follow-up time after the fracture. In the same report, women made up the majority of the 102 patients treated, BPs being the most used agents.

A study evaluated persistence and adherence to treatment in 59,782 patients with hip fracture [38]. Only 58% received bisphosphonate treatment, 6.7% identified as compliant with MPR (medication possession ratio, sum of days of supply divided by 365 days) higher than 80%, and 12.3% were persistent users (length of time with continuous therapy without a refill gap longer than 30 days in 12 months) [38]. Only 35% of the patients refilled their first prescription after a hip fracture [14].

A large study [50] evaluated anti-osteoporotic treatment in 51,346 hip fracture patients admitted to 318 hospitals in the U.S.A and showed that only 7.3% of patients received specific anti-osteoporosis treatment after the fracture and only 2% received additional calcium and vitamin D, with no difference in the odds of receiving related to age groups.

The APROP clinical trial (clincaltrials.gov identifier NCT03044015, study completion date in December 2021) aims to offer data regarding the impact of primary care interventions on the maintenance of the adherence to the pharmacological therapy. A sub-study of the Norwegian Capture the Fracture Initiative (clinicaltrials.gov Identifier NCT02608801) aims to predict second fractures, one of the second outcomes being self-reported adherence to treatment, with no published results in March 2020.

## 4. Mortality

A low bone mass density (BMD) with or without a rapid loss of bone mass was associated with increased mortality risk [51,52]. However, the mechanisms are uncertain considering that, to date, no bone-related factor vital for bone catabolism was demonstrated to affect survival or, inversely, a bone factor that lowers mortality risk independent of fracture risk reduction. An association between a low BMD and a significant risk of cardiovascular mortality and all-cause mortality risk was analyzed [53]. An increased coronary artery calcification seems to correlate with a lower BMD. Both factors are independently associated with the severity of artery calcification on the coronaries with the power to predict mortality [54].

Encouraging news about the anti-osteoporotic treatment being associated with lower mortality rates in osteoporotic patients has been published, and the authors showed increased survival in treated patients [16,27]. A meta-analysis of 61 randomized controlled trials showed a lower risk of cardiovascular mortality (RR = 0.81, 0.64–1.02, in 10 studies) and a significant reduction in all-cause mortality (RR = 0.90, 0.84–0.98 in 48 studies) with BP treatment [16].

At the same time, a recent comprehensive meta-analysis of randomized placebo-controlled clinical trials published in 2019 [55] suggested that anti-osteoporotic treatment and bisphosphonates particularly were not associated with lower mortality rates. The majority of the included studies in the meta-analysis reported only osteoporotic patients without a fragility fracture [55].

Our review includes 21 articles published in the last ten years that analyzed the effect on mortality after hip fracture in patients on anti-osteoporotic treatment, before or after the index fracture. Most studies examined the impact of bisphosphonates. Two studies reviewed zoledronic acid solely, and one analyzed together oral (oBP) and intravenous (iBP) bisphosphonates and denosumab. Five of the studies were retrospective; three of them included a large number of patients. All of the reviewed studies showed a lower mortality rate after hip fracture in patients receiving anti-osteoporotic treatment.

A randomized prospective study published in 2011 [16] showed an 8% (adjusted hazard ratio of 0.92 per month treated) relative mortality rate reduction per month and a 63% reduction per year of treatment. Although the authors recognized the limitations of analyzing a small number of a relatively young and healthy cohort with hip fractures, the results were almost too good to be true. BP treatment would have achieved a nearly 100% reduction in mortality [16].

Research from 2013 [14] and 2019 [15] showed that a lower mortality rate in BP-treated hip fracture patients raised the question of possible confounders. These can be attributed in part to the survival benefit observed after treatment, but at the same time, the existence of bias does not rule out a possible actual effect of treatment with BP. The first study [14] showed lower 3-month mortality in BP-treated patients before fracture (OR 0.68, CI 95% 0.59–0.77, OR 0.73, CI 95% 0.61–0.88) for treatment initiation after the fracture, with lower risk observed even in patients who filled only one prescription (OR = 0.84, CI 95% 0.73–0.95). The difference in risk reduction, from 16% to 27%, might have been related to the BP treatment, but the association was not demonstrated by the authors [14]. Moreover, the tendency of a decreased mortality trend with continued prescription refilling was not statistically significant when adjusted for comorbidities and comedications [14]. Simultaneously, in the randomized control trial that showed decreased mortality with zoledronic acid, the effect was not observed for at least 12 months [12].

In the second study [15], the authors examined hip fracture mortality in patients provided by the Swedish pharmacies databases and showed a 15% lower mortality (HR 0.85, 0.79–0.91) in BP-treated patients. This risk was lower starting from day 6 of treatment. The median follow-up of 2.8 years with a mortality of 29% was similar to other studies. Oral alendronate was the most prescribed medication, followed by a small fraction of risedronate and zoledronic acid. No particular group of patients showed a higher reduction in mortality rate [15].

A 2019 study first analyzed the effect of denosumab on survival in hip fracture patients [13]. The authors [13] compared anti-osteoporotic treatment and naive patients in terms of mortality and refracture and showed a decreased risk of death associated with denosumab treatment (26%, HR 0.74, CI 95% 0.58–0.94, *p* < 0.05), oBP (30%, HR 0.70, CI 95% 0.62–0.7, *p* < 0.05) and iBP (37%, HR 0.63, CI 95% 0.55–0.71, *p* < 0.001). In Cox regression analysis, only women taking iBP were associated with a lower mortality risk (17% reduction risk, HR 0.83, CI 95% 0.71–0.98, *p* = 0.023), but for denosumab and oBP, the results were not statistically significant [13].

The lowest hazard ratio for mortality at one year after the event was found in a retrospective study that included more than 30,000 patients [17] (HR of 0.43, CI 95% 0.36–0.52) in patients with BP treatment started after the index hip fracture. The same risk for male patients was 0.59 (CI 95% 0.39–0.88, *p* < 0.05). Before the index fracture, BP-treated patients in the 50–69 age group demonstrated higher odds for mortality compared to non-treated patients. Male patients aged 85 and above, treated with iOR, showed reduced odds of mortality at one year after the hip fracture (HR 0.50, CI 95% 0.27–0.96, *p* < 0.05) [17]. Another study included in the review, with a similar study design, showed lower overall mortality regardless of the treatment initiation time, before or after the index fracture [14]. In BP-treated women before the index fracture, lower rates of mortality were observed at one year (OR 0.79, CI 95% 0.7–0.89, *p* < 0.0001) and three years (OR 0.78, CI 95% 0.7–0.86, *p* < 0.05) after the index fracture [17].

One of the meta-analyses included in the study encompassed four trial studies, two randomized and two non-randomized, prospective matched controlled studies, one being the HORIZON Recurrent Fracture Trial [12]. Two articles out of four (1447 hip fracture patients) directly evaluated mortality in the BP and the control group, and in the other two, mortality data were extracted. There was a mean significant difference of 0.66 (CI 95% 0.52–0.85, *p* = 0.001) but with no statistical heterogeneity (I^2^ = 0%, *p* = 0.52). Simultaneously, other complications besides mortality and the risk of second hip fractures were more common in the BP-treated group than placebo (mean significant difference of 1.3, CI 95% 1.10–1.54, *p* = 0.002, also with low heterogeneity I^2^ = 0%, *p* = 0.57). A particularity of this meta-analysis [12] is the inclusion of non-randomized prospective studies, explained by the insufficient randomized data available. The lack of compliance data can influence the results but the authors stated that it was higher than 80% in all four studies [12]. Taking this into consideration, as well as the scarcity of data in the literature, this meta-analysis [12] offers important insights regarding mortality and also re-fracture risk in treated hip fracture patients.

In the analysis of 102 patients who underwent hemiarthroplasty after hip fracture, the mean survival rate was 30.2–30.7% for non-treated patients and 33.3% for treated patients, with a 5-year survival rate 16% for BP-treated patients compared to 5% for non-treated ones (*p* = 0.002) [19].

The implementation of FLS did not show an improved mortality rate in already treated patients, only when compared to non-treated hip fracture individuals before FLS [32].

One of the most extended follow-up periods of the analyzed studies, namely eight years, was found in van Geel et al. [20], with a lower mortality rate for BP-treated patients (HR 0.79, 0.64–0.97). Although the study design included all major fragility fractures, not only hip, the outcomes were adjusted for age, BMD, and initial fracture severity, and the survival benefit was not dependent on the relative risk reduction of subsequent fractures [20].

The most significant number of patients covered was 163,273 (all major osteoporotic fractures) in the study of Abtahi et al., recently published in 2020. It showed a 28% lower mortality after hip fracture in current BP-treated patients and, interestingly, a 42% lower mortality after hip fracture in patients with past BP exposure (>1 year) [22] using Cox proportional hazard models.

The HORIZON Zoledronic Acid Once-Yearly Recurrent Fracture Trial One was one of the first studies demonstrating a lower mortality rate after hip fracture associated with anti-osteoporotic treatment [12]. It investigated zoledronic acid [12], with findings that led to the European Union’s approval of zoledronic acid treatment in osteoporosis. Other studies published in 2011 [23] and 2014 [24] performed secondary analyses using the patients database from the HORIZON study. The first study mentioned [23] showed lower mortality after 5 mg of zoledronic (HR 0.71, 0.46–1.31 in men and HR 0.74, 0.54–1.02 in women), but with a short median follow-up of 1.9 years. In the second study [24], the authors analyzed zoledronic acid’s effect in a subgroup of cognitively impaired patients. A minor difference in mortality rate between the two groups in cognitively impaired patients (23.2% compared to 26.9%) was observed, as well as lower mortality rates in the treatment arm (6.2% compared to 10.5% in the placebo arm, *p* < 0.001). Another secondary analysis of the HORIZON study [56] showed a lower relative risk of pneumonia or a lower respiratory infection associated with bisphosphonate therapy, although not statistically significant (*p* > 0.05). Risk and mortality after pneumonia were significantly lower in bisphosphonate-treated patients compared to naive patients but also when compared to other anti-osteoporosis treatments [57]. Alendronate was associated with lower risk of cardiovascular death (HR of 0.33, *p* = 0.001) and myocardial infarction (HR of 0.55, *p* = 0.014) after 1 year of treatment in a retrospective cohort study [25].

A clinical trial [58] aimed to evaluate the effects of geriatric co-management in hip fracture patients. Although no results are published, the objectives set by the authors aim to evaluate prospective effects of the Geriatric Fracture Centers, on hip fracture patients.

Teriparatide was the pro-osteogenic agent investigated. Most of the studies regarding teriparatide and hip fracture are related to secondary outcomes such as accelerated fracture healing and union, with less data regarding mortality or subsequent fractures [29]. The meta-analysis regarding the effect of teriparatide included in the review showed data from two clinical trials and three retrospective cohort studies [29], with a total of 607 patients. The model showed no significant effect on mortality or subsequent fracture risks [29]. The most important limitation is the lower heterogeneity between included studies, related to the inconsistencies between treatment doses and duration [29].

The relationship of mortality risk secondary to a decreased subsequent re-fracture risk was negated in a small study of 325 patients [27].

One of the main limitations of the included studies was the missing information regarding treatment compliance or adherence. As discussed previously, compliance and adherence to anti-osteoporotic treatment are significant healthcare problems, both in quantifying its magnitude in clinical practice and in implementing the necessary measures to decrease the treatment gap.

The most valuable initiation time for anti-osteoporotic treatment (before or after the index fracture) in terms of lower mortality rates is not precise, with studies such as Behanova et al. [13] and Brozek et al. [17] showing that initiation after the index fracture is more effective than before the fracture. Although the studies presented included large numbers of patients, perhaps because of the low adherence and compliance, even more significant numbers are needed to validate the actual results.

The mechanism through which the BP treatment influences mortality is still poorly understood. The effects on the accelerated bone turnover [59], the anti-inflammatory effect [60], or the modulation of pro-inflammatory cytokines [61] have been stipulated as possible mechanisms.

## 5. Second Fracture

It is well known that a prevalent fragility fracture almost doubles the risk of a second fracture [2]. The high mortality associated with hip fracture is even higher in patients who suffer recurrent fractures [62]. A link between the lower mortality after hip fracture in patients with anti-osteoporotic treatment and subsequent lower risk of re-fracture was searched and investigated.

Our review included 20 articles published in the last ten years that analyzed the effect on re-fracture rate after hip fracture in patients on anti-osteoporotic treatment, before or after the index fracture.

One meta-analysis [18] scrutinized the associated risk of a second hip fracture in BP-treated and in a control group of hip fracture patients (3088 cases) and showed a significant mean difference of 0.60 (CI 95% 0.39–0.93, *p* = 0.02) also without statistical heterogeneity (I^2^ = 35%, *p* = 0.20). Only one study [55] was statistically significant, and two showed higher risk for treated patients. The 80% higher compliance present in all the included studies further emphasizes the effect found. The second analysis derived from the HORIZON Zoledronic Acid Once-Yearly Recurrent Fracture Trial [23] found the rate of a new fracture after treatment with zoledronic to be 8.9% in treated women compared to 15.6% in the placebo group, *p* < 0.001. Although the absolute numbers were similar for men, the association was not statistically significant [23].

After eight years of follow-up, a prospective cohort study found a lower risk for subsequent fractures (HR 0.60, 0.49–0.73) in treated patients included in Fracture Liaison Services (FLS). Other medical interventions included in addition to anti-osteoporotic treatment can explain in part this association [20]. The FLS implementation was not correlated with a decreased risk of second fractures [32]. All new fractures were recorded in patients not treated in a group multicenter prospective study that investigated the results of implementing FLS in Greece [40].

Treatment with risedronate proved efficient in preventing subsequent contralateral hip fracture in Japanese women with unilateral hip fracture (HR 0.31, 0.12–0.79 in univariate and 0.218, 0.074–0.63 in multivariate analysis for subsequent fracture risk in risedronate-treated patients) [36] in a small group.

The only study that analyzed treatment exclusively with denosumab [35] showed a 39% relative risk reduction of a secondary fragility fracture in denosumab-treated patients (10.5% compared to 17.3% in the non-treated group, *p* < 0.0001) in 7808 patients.

In a relatively young and healthy small hip fracture cohort, additional fractures after an index hip fracture were observed in 53% without treatment and 26% BP-treated patients before the event [16].

In the analysis of hemiarthroplasty-only-treated patients, the authors did not observe a significant difference in the risk of subsequent fractures in treated and non-treated patients [19].

In a nationwide population-based observational study, in 87,145 patients, BP treatment after hip fracture had a negative risk correlation with second fracture (20.8% versus 32.3%, *p* = 0.023), adjusted OR = 2.24, 1.38–2.90, *p* = 0.017 [34] for 7-year follow-up after the index fracture. Other anti-osteoporotic agents were not observed to have a role in second fracture prevention [34].

Another population-based cohort study [39] that included 88,320 hip fracture patients found a statistically significant correlation between re-fracture rate and alendronate treatment, also related to the medication possession ratio (MPR). Only 10.5% of patients were identified as users and only 1.68% had the highest MPR, between 75 and 100% (10). High MPR was associated with a protective second fracture relative risk (HR 0.66, 0.49–0.88, *p* < 0.005 for 50–75% MPR and HR of 0.61, 0.47–0.78, *p* < 0.001 for MPR 75–100%) (10). Interestingly, although not statistically significant, MPR higher than 50% was actually a risk for re-fracture [39]. Although MPR is a good surrogate to evaluate treatment, the compliance is not optimally evaluated. At the same time, alendronate is less recommended for severe osteoporosis, as hip fracture patients qualify. A retrospective cohort study using the same national database as the previously mentioned paper [30] showed that persistent anti-osteoporosis treatment use was associated with a lower fracture risk in 946 patients (HR 0.64, 0.41–0.99, *p* = 0.043), persistence being defined as continuous refills with specific end points for discontinuation specific for every class or sub-class of agents.

All studies found a negative association between mortality after hip fracture and concomitant anti-osteoporotic treatment. Nevertheless, the data regarding subsequent fracture risk were contradictive. In the study of Behanova et al.[13], higher risk of subsequent hip fracture in patients with antiresorptive treatment was observed in men with oBP (HR 2.89, 1.58–5.30), women on DMAB (HR 1.77, 1.08–2.91), and iBP (HR 1.81, 1.35–2.41), and, surprisingly, the rate of subsequent hip fractures did not decrease over time. In addition, in another study, higher hip re-fracture rates were found in BP-treated patients, before or after the index hip fracture, regardless of age, with the highest OR 1.87 (CI 95% 1.32–2.66) of the second fracture in BP users aged 70–84, after 1-year post-fracture [17].

The authors of the mentioned studies stated that a worse osteoporotic status could explain the higher rate of re-fracture in treated patients, which is the cause of the higher re-fracture rates.

Although new data showed that anti-osteoporotic treatment does not have an effect on mortality in osteoporotic patients [55], data regarding only hip fracture patients are still scarce and insufficient. The excess mortality in hip fracture patients [63] compared to osteoporotic individuals without fracture is an important argument to continue the research for possible beneficial effects for this category. The increased trends of hip fracture incidence worldwide [64] and the relatively unchanged mortality rates in the last few decades [5] in hip fracture patients further necessitate the search for possible treatment benefits.

The effect of re-fracture can be related to the increased mortality but also other outcomes, such as increased dependency and decreased quality of life, in hip fracture patients.

The implementation of fracture-dedicated services such as the “Capture the Fracture Campaign” and “Fracture Liaison Services” through “Care Manager Programs” showed promising results in reducing the subsequent fracture rate and mortality and also managed to increase the percentage of patients receiving specific anti-osteoporosis treatment [1]. Inclusion of primary care and ortho-geriatric assessment and monitoring can increase the percentage of treated patients, adherence, and compliance. Since regionality, religion, and socioeconomic status can translate into a geographic variation in treatment adherence and compliance, larger, randomized prospective studies are needed to validate the actual results.

## 6. Conclusions

Most of the studies showed notable effects on mortality and re-fracture rates associated with anti-osteoporotic treatment, mostly for oral and intravenous bisphosphonates. Larger, randomized prospective studies are needed to validate the actual results.

## Figures and Tables

**Figure 1 jpm-11-00341-f001:**
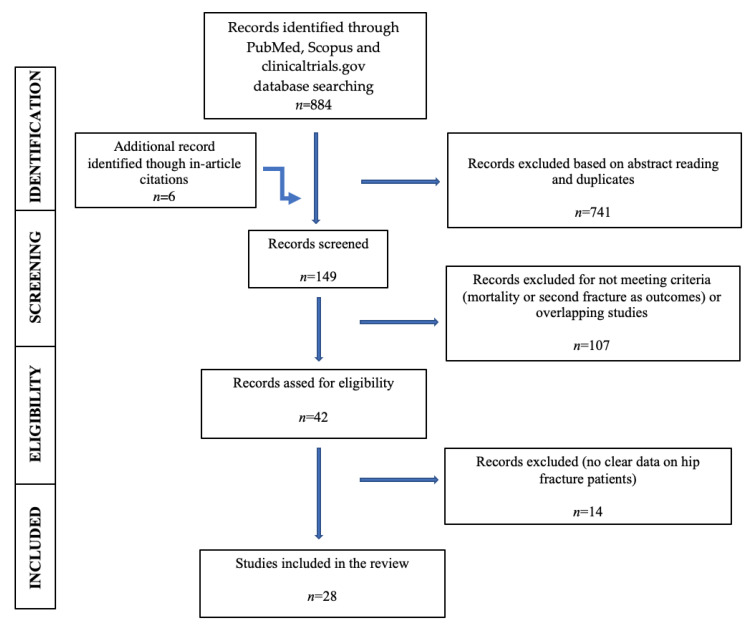
PRISMA flow diagram.

**Table 1 jpm-11-00341-t001:** Descriptions of the studies with main findings and limitations.

Authors/Year	Agent Investigated	Study Type	*n*	Main Results	Limitations
***Mortality***					
Behanova et al., 2019 [13]	oBP iBP DMAB	retrospective	54,145	- 17% lower risk of dying in treated women (HR 0.83, 0.71–0.98, *p* = 0.023) statistically significant only for intravenous bisphosphonates	- lack of criteria associated with treatment prescription - no data regarding compliance with treatment
Bondo et al., 2013 [14]	BP	cohort study	42,076	- lower 3-month mortality in BP-treated patients before fracture (OR 0.68, 0.59–0.77, OR 0.73, 0.61–0.88) for treatment initiation after the fracture	- possible confounders: lower mortality even in patients who filled only one prescription (OR = 0.84, CI 95% 0.73–0.95)
Bergman et al., 2019 [15]	BP	retrospective, cohort	49,765	- 15% lower mortality (HR 0.85, 0.79–0.91) in BP-treated patients	-possible confounders: the risk was lower starting from day 6 of treatment
Sambrook et al., 2011 [16]	BP	randomized prospective	220	- BP treatment was associated with reduced mortality after hip fracture (aHR of 0.92 per month treated) 8% relative reduction per month and 63% per year of treatment	-relatively young and healthy hip fracture cohort -small number of patients
Brozek et al., 2016 [17]	BP	retrospective	31,668	- lower risk of mortality with BP treatment before and after the hip fracture (lowest HR of 0.43 for treatment started after hip fracture)	-no data regarding compliance to treatment
Peng et al., 2016 [18]	BP	metanalysis of 4 trials (2 randomized and 2 prospective matched controlled studies)	3088	- lower mortality risk in BP-treated patients (OR 0.66, 0.52–0.85, *p* = 0.001)	- few studies, mixed randomized and non-randomized studies no data regarding the type of BP, dose, duration - no statistical heterogeneity (I^2^ = 35%) -dominated results by the HORIZON Recurrent Fracture Trial [12]
Cobden et al., 2019 [19]	BP	retrospective multicenter study	562	- 5-year survival rate was 16% for BP-treated patients compared to 5% for non-treated ones (*p* = 0.002);	- retrospective study no predictive risk factors for mortality included in the analysis - only patients after hemiarthroplasty were investigated -small number of patients
van Geel et al., 2018 [20]	BP	prospective cohort study	5011 (2534 on BP)	- lower mortality rate for BP treated patients (HR 0.79, 0.64–0.97) in 8 years of follow-up	- no data regarding adherence to treatment -patients included in the Fracture Liaison Services with more medical interventions included than anti-osteoporosis treatment -all major fracture included
Wang et al., 2019 [21]	BP	retrospective	690	- lower mortality for BP and non-BP osteoporosis medication (HR 0.35, 0.19–0.64) and (HR 0.49, 0.34–0.69)	- retrospective study -only 57% of the fractures were followed -small number of patients
Abtani et al., 2020 [22]	BP	population-based cohort study	163,273	- 28% lower mortality after hip fracture in current BP-treated patients - 42% lower mortality after hip fracture in patients with past BP exposure	- no data regarding adherence to treatment
Boonen et al., 2011 [23]	Zoledronic Acid	randomized, placebo-controlled, double-blind trial	508 men (248/260) 1619 women	- lower mortality after 5 mg of Zoledronic (HR 0.71, 0.46–1.31 in men) and (HR 0.74, 0.54–1.02 in women)	- short median follow-up of 1.9 years
Prieto-Alhambra et al., 2014 [24]	Zoledronic Acid	secondary analysis from HORIZON randomized controlled trial	4093 1966/2127	- lower mortality rates in the treatment arm (6.2% compared to 10.5%, in the placebo arm, *p* < 0.001) - smaller difference in mortality rates between the two groups in cognitive impaired patients (23.2% compared to 26.9%)	- study design with main focus on cognitive impaired patients
Sing et al., 2018 [25]	BP	retrospective cohort study	4594/13,568	- lower cardiovascular mortality rate (HR 0.33, 0.17–0.65) for alendronate and (HR 0.35, 0.2–0.63) for other BPs at 1 year -the lower realtive risk is mantained up to ten years of follow-up for alendronate (HR 0.59 0.44–0.79, p) or other BPs (HR 0.58, 0.44–0.75)	- no clear criteria for treatment recommendation - only one prescription needed for inclusion - possible confounding cardiovascular risk factor not included
Nordstrom et al., 2017 [26]	BP	restropective cohort study	5845/15,518	- decreased risk of death adjusted for all covariates (HR 0.79, 0.73–0.85)	-observational study -high mean time between the hip fracture and initiation of BP (331 days, rage of 1 to 2770
Center et al., 2011 [27]	BP	prospective cohort Dubbo study	1223 women/819 men (429 with fractures)	-lower risk of mortality in women (HR 0.33, 0.16–0.66) in the multivariate analysis	-no additional data regarding fracture type or the number of hip fractures included -observational study - lack of treatment recommendation criteria
Degli Esposti et al., 2012 [28]	Mainly BP	retrospective cohort study	5636 (187 pre-fracture/651 post fracture)	- 62.3% lower risk of death (−73.4%–−46.6%) in treated patients, (HR 0.377, 0.266–0.534, *p* < 0.001)	-retrospective study with small number of patients -common analysis for all anti-osteoporosis agents -all hip fractures regardless of traumatic event
Han et al., 2020 [29]	Teriparatide	metanalysis of 6 studies (2 randomized and 4 retraospective studies)	607 (269/338)	- lower mortality in the teriparatide group in the fixed model (OR 0.34, 0.13–0.88, *p* = 0.03) but the random-effect was not significant (OR 0.37, 0.12–1.09, *p* = 0.07)	- no heterogeneity (I^2^ of 4%) -observational studies that can exaggerate the effects -inconstant treatment duration -confounding factors not assessed
Hsu et al., 2020 [30]	BP/Raloxifene/Teriparatide/Denosumab	retrospective cohort study	946 (210/736)	- lower risk of mortality in the persistence group (HR 0.83, 0.62–1.11), *p* = 0.21 for all anti-osteoporosis treatment - higher risk of mortality in patients aged 70–79 (aHR = 1.29, 1.08–4.86, *p* = 0.031) and >80 (aHR = 3.11, 1.49–6.5, *p* = 0.003) in persistence group	- small number of patients - no separate analysis for the non-BP agents - certain confounding factors not assessed
Lebanon et al., 2020 [31]	Zoledronic acid (46.3%)/Denosumab (34.1%)/Teriparatide	prospective cohort study	253 (85/168)	-mortality rate in treated patients was 5.1% at one year compared to 26.3% in naive patients (*p* < 0.001)	- possible effects of calcium and vitamin D - small number of patients
Gonzalez-Quevedo et al., 2020 [32]	any treatment	prospective cohort study	724	-all-cause mortality was lower in the FLS-treated group (HR 0.66, 0.47–0.94) compared to all patients before FLS - treatment after the implementation of the FLS associated higher mortality risk compared to treatment initiated before (1.75, 0.54–5.49)	- small number of patients - no cause-and-effect relationtioship between mortality and FLS - small number of patients - patients included in the FLS with more medical interventions included than anti-osteoporosis treatment
Rotman-Pikielny et al., 2018 [33]	any treatment	prospective cohort study	218/219	- lower mortality rates in treated patients (4.3% versus 21.8%) with a 53% decrease in mortality risk in female sex (HR 0.47, 0.30-0.72, *p* < 0.001) -	- small number of patients - outcomes observed in a multidiscliplinary team with probability of secondary unrelated effects in compliant patients - confounding factors not assessed
**Second Fracture**					
Shen et al., 2014 [34]	BP	nationwide population-based longitudinal observational study	87,415	- BP treatment after hip fracture had a negative risk association with second fracture (20.8% versus 32.3%, *p* = 0.023), aOR = 2.24, 1.38–2.90, *p* = 0.017	- register-based study
Palacios et al., 2015 [35]	DMAB	randomized post hoc analysis FREEDOM	7808	- 39% relative risk reduction for a secondary fragility fracture in denosumab treated patients (10.5% compared to 17.3% in the non-treated group, *p* < 0.0001)	- post hoc analysis
Behanova et al., 2019 [13]	oBP iBP DMAB	retrospective	54,145 1919 oBP 1870 iBP 555 DMAB 42,795 untreated	- higher risk of subsequent hip fracture in patients with antiresorptive treatment: men with oBP (HR 2.89, 1.58–5.30), women on DMAB (HR 1.77, 1.08–2.91), and iBP (HR 1.81, 1.35–2.41)	- lack of data regarding adherence to treatment - short follow-up period - data regarding criteria for treatment recommendation
Sambrook et al., 2011 [16]	BP	randomized prospective	220	- additional fractures were observed in 53% without treatment and 26% BP treatment before the event	- relatively young and healthy hip fracture cohort -small number of patients
Brozek et al., 2016 [17]	BP	retrospective	31,668	- higher hip refracture rates in BP treated patients, before or after the index hip fracture, regardless of age - OR 1.87, 1.32–2.66 of second fracture in BP users age 70–84, after 1-year post fracture	- no data regarding compliance to treatment
Peng et al., 2016 [18]	BP	metanalysis of 4 trials (2 randomized and 2 prospective matched controlled studies)	3088	- second hip fracture different rate between the BP group and the control group (mean difference of 0.6, 0.39–0.93, *p* = 0.02)	- few studies, mixed randomized and non-randomized studies no data regarding the type of BP, dose, duration - no statistical heterogeneity (I^2^ = 35%)
Cobden et al., 2019 [19]	any treatment	retrospective multicenter study	562	- no significant differences in second fracture risk	-small number of patients - only patients after hemiarthroplasty were investigated
van Geel et al., 2018 [20]	BP	prospective cohort study	5011 (2534 on BP)	- lower risk for subsequent fractures (HR 0.60, 0.49–0.73)	- no data regarding adherence to treatment -patients included in the Fracture Liaison Services with more medical interventions included than anti-osteoporosis treatment -all major fracture included
Osaki et al., 2012 [36]	Risedronate	prospective matched cohort study	529 (173/356)	-HR 0.31, 0.12–0.79 in univariate and 0.218, 0.074–0.63 in multivariate analysis for subsequent fracture risk in risedronate treated patients	-no randomization -history of BP treatment in the control group -small number of patients
Liu et al., 2019 [37]	Zoledronic Acid	randomized controlled trial	482 (353/129)	- lower refracture rate in the treatment group (5.9% compared to 8.5%, *p* < 0.01)	-short observation time -small number of patients
Prieto-Alhambra et al., 2014 [24]	Zoledronic Acid	secondary analysis from HORIZON randomized controlled trial	4093 1966/2127	- no significant correlation between treatment arm and placebo arm regarding re-fracture (*p* > 0.05)	- study design with main focus on cognitively impaired patients
Lee et al., 2013 [38]	BP	retrospective	59,782	- lower incidence of second hip fracture in compliant patients compared to non-compliant (0.8% versus 2.3%, *p* < 0.001) - lower incidence of second hip fracture in persistent users compared to non-persistent (0.9% versus 2.4%, *p* < 0.001)	-retrospective study -all hip fractures included regardless of the traumatic event
Boonen et al., 2011 [23]	Zoledronic Acid	randomized, placebo-controlled, double-blind trial	508 men (248/260) 1619 women	- 8.9% rate of new fractures in zoledronic acid treated compared to 15.6% in placebo-treated women	- short median follow-up of 1.9 years
Nordstrom et al., 2017 [26]	BP	restropective cohort study	5845/15,518	- after BP initiation, the risk of hip fracture was lower (HR 0.76, 0.65–0.90) - BP users of more than 90 days had a lower risk of new hip fracture (HR 0.69, 0.54–0.87) - effect on second hip fracture risk decrease was seen even in later BP users with OR of 2.22 compared to 2.63 in never users - a decrease for any subsequent fracture was seen (HR 0.90, 0.78–1.04)	-observational study - new fracture data based on the national registry entries -high mean time between the hip fracture and the initiation of BP (331 days, rage of 1–2770)
Han et al., 2020 [29]	Teriparatide	metanalysis of 6 studies (2 randomized and 4 retraospective studies)	607 (269/338)	- no significant difference in subsequent fracture risk between the two groups (OR 0.60, 0.30–1.18, *p* = 0.14)	- no heterogeneity (I^2^ of 0%) -observational studies that can exaggerate the effects -inconstant treatment duration -confounding factors not assessed
Degli Esposti et al., 2012 [28]	Mainly BP	retrospective cohort study	5636 (187 pre-fracture/651 post fracture)	- lower risk of re-fracture of -53.3% (−67.3% – −33.2%), *p* < 0.001	-retrospective study with small number of patients -common analysis for all anti-osteoporosis agents -all hip fractures regardless of the traumatic event
Hsu et al., 2020 [30]	BP/Raloxifene/Teriparatide/Denosumab	retrospective cohort study	946 (210/736)	- lower rate of recurrent fractures in the persistence group for all agents (aHR 0.64, 0.49–0.99, *p* = 0.043) -persistence BP users with lower recurrent fracture rates (aHR 0.54, 0.32–0.90, *p* = 0.018)	- small number of patients -no separate analysis for the non-BP agent- certain confounding factors not assessed
Gonzalez-Quevedo et al., 2020 [32]	any treatment	prospective cohort study	724	- no statistically significant difference between groups	- small number of patients;- no cause-and-effect relationtioship between mortality and FLS - small number of patients - patients included in the FLS with more medical interventions included than anti-osteoporosis treatment
Chen et al., 2020 [39]	Alendronate	population-based cohort study	88,320 (9278/79,042	-lowest risk of second fracture in the MPR 75–100% (aHR 0.61, 0.47–0.78, *p* < 0.001)	- certain confounding factors not assessed - incomplete treatment adherence data - lack of criteria choice for aledronate treatment
Makras et al., 2020 [40]	any treatment	multicenter prospective study	392	- significant increase in new fractures in patients not receiving anti-osteoporosis treatment, *p* < 0.001	-small number of patients -small period of treatment- effects possibly related to the FLS interventions besides treatment

*n*—number of patients, BP—bisphosphonates; oBP—oral bisphosphonates; iBP—intravenous bisphosphonates; DMAB—denosumab; HR—hazard ratio; aHR—adjusted hazard ratio; OR—odds ratio, aOR—adjusted odds ratio; FLS—Fracture Liaison Service; MPR—medication possession ratio.

## Data Availability

All information included in this review is documented by relevant references.

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
