# Peer review of "Adherence to Anti-Osteoporotic Treatment and Clinical Implications after Hip Fracture: A Systematic Review"

_jpm, 2021, doi:10.3390/jpm11050341_

Round 1
Reviewer 1 Report
The authors should consider the followings:
- In the database searching in clinical studies, the authors may include more databases, such as clinicaltrials.gov. Please also explain why the authors only choose pubmed.
- In Figure 1, from Screening to Eligibility, and from Eligibility to Included, the authors should include the inclusion criteria in the flow chart.
- The authors may add a summary to summarize the limitation included (but not limited) in Table 1.
- The authors should include the suggestion of future direction(s) in the final paragraphs.
- The authors should also comments on the pro-osteogenic agent, if any.
- In the conclusion, by “bisphosphonates” did the authors suggested all types of bisphosphonates. Or just a few sub-classes?
- As a review article, the group should cover more relevant references.
- The authors should consider using English proof-reading services by language professional.
- In Figure 1, the authors should list clearly the database used, in the flow chart.
- The authors should give rationale(s) of why they still continue the review with only 17 studies included.
- Please give rationale, why the authors did not cite this article from 2019, Yu, SF., Cheng, JS., Chen, YC. et al. Adherence to anti-osteoporosis medication associated with lower mortality following hip fracture in older adults: a nationwide propensity score-matched cohort study. BMC Geriatr 19, 290 (2019). https://doi.org/10.1186/s12877-019-1278-9
Author Response
Title: Cover Letter
Re-submission of manuscript ID: jpm-1191565
April 18, 2021
Dear Reviewer-1,
Thank you for the opportunity to revise and resubmit our manuscript entitled “Adherence to anti-osteoporotic treatment and clinical implications after hip fracture. A systematic review” (manuscript ID: jpm-1191565). Please find attached the revised version of the original article that we would like to resubmit for consideration and publication in the Journal of Personalized Medicine. We appreciate all the careful reviews and constructive suggestions from the Reviewer-1. We believe that the manuscript is substantially improved after making the suggested edits and changes.
The point-by-point responses to the Reviewer’s comments and the revision details are provided, indicating how and where the main text or Table were changed. Track changes highlight all of the modifications we made in the original paper so that they can be seen as red or green, underlined text within the manuscript, or in small bubbles on the left of the text whenever a deletion was made.
Finally, we would like to declare; that this revision has been developed in close consultation with all co-authors, and each author approved the final, revised form of this manuscript. We look forward to hearing from you regarding our re-submission, and we are open to respond to any further questions or comments you may have.
Thank you for your consideration of this manuscript!
Sincerely yours,
Ramona Dobre, MD, PhD student
Corresponding author
RESPONSE TO REVIEWER-1 COMMENTS
We want to thank the Reviewer for the careful and thoughtful review of our manuscript and for all of the valuable suggestions he/she made. We hope that we satisfactory responded to all of the comments and made the necessary changes to the manuscript. Thank you once again for taking the time and energy to help us to improve the quality of our paper.
Point 1: In the database searching in clinical studies, the authors may include more databases, such as clinicaltrials.gov. Please also explain why the authors only choose pubmed.
Response 1: Thank you for your suggestion. Clinicaltrials.gov and also Scopus were included in the database searching.
Point 2: In Figure 1, from Screening to Eligibility, and from Eligibility to Included, the authors should include the inclusion criteria in the flow chart.
Response 2: Changes were made to Figure 1 and Materials and Methods. Exclusion criteria were included in Figure 1 and Materials and Methods was updated with the used inclusion criteria (lines 87-92).
Point 3: The authors may add a summary to summarize the limitation included (but not limited) in Table 1.
Response 3: Thank you for your suggestion. We added a Supplementary Table 1 with data regarding study limitations.
Point 4: The authors should include the suggestion of future direction(s) in the final paragraphs.
Response 4: Thank you for your thoughtful remark. Future directions were added (lines 928-945).
Point 5: The authors should also comment on the pro-osteogenic agent, if any.
Response 5: Thank you for your suggestion. We added comments for Teriparatide (in the Mortality section, lines (749-756) and Table 1).
Point 6: In the conclusion, by “bisphosphonates” did the authors suggested all types of bisphosphonates. Or just a few sub-classes?
Response 6: Thank you for your question. Since most of the included studies did not specify additional data regarding sub-classes, the statement refers to the whole class. The conclusion section was updated.
Point 7: The authors should consider using English proof-reading services by language professional.
Response 7: Thank you for your observation; the manuscript was corrected by a colleague who is a Native English speaker. Furthermore, we checked the whole text with a Writing Assistant Program to eliminate all of the remaining grammar mistakes.
Point 8: In Figure 1, the authors should list clearly the database used, in the flow chart
Response 8: Figure 1 was modified accordingly, adding Scopus and clinicaltrials.gov databases in the Flow chart.
Point 9: The authors should give rationale(s) of why they still continue the review with only 17 studies included.
Response 9: Based on your suggestion, we included more databases and studies in our review.
Point 10: Please give rationale, why the authors did not cite this article from 2019, Yu, SF., Cheng, JS., Chen, YC. et al. Adherence to anti-osteoporosis medication associated with lower mortality following hip fracture in older adults: a nationwide propensity score-matched cohort study. BMC Geriatr 19, 290 (2019). https://doi.org/10.1186/s12877-019-1278-9
Response 10: Lastly, but not least important, we thank you once again for the suggestions that came from a meticulous read of our material. The above-mentioned study was included.

Reviewer 2 Report
Unique original and innovative meta analysis into review and adherence of osteoporosis treatment post hip fracture and examining mortality rates.Good descriptive meta analyses with limited papers that could be attributed in this area of research.
Good Introduction of problem
Appropriate description of methods and materials and how studies were collaborated and studied
Accurate and comprehensive conclusion of individual studies
Need to integrate meta analysis more comprehensively in a more accurate conclusion
Need to improve grammatical English and more thorough and comprehensive conclusion of findings
Author Response
Title: Cover Letter
Re-submission of manuscript ID: jpm-1191565
April 18, 2021
Dear Reviewer-2,
Thank you for the opportunity to revise and resubmit our manuscript entitled “Adherence to anti-osteoporotic treatment and clinical implications after hip fracture. A systematic review” (manuscript ID: jpm-1191565). Please find attached the revised version of the original article that we would like to resubmit for consideration and publication in the Journal of Personalized Medicine. We appreciate all the careful reviews and constructive suggestions from the Reviewer-2. We believe that the manuscript is substantially improved after making the suggested edits and changes.
The point-by-point responses to the Reviewer’s comments and the revision details are provided, indicating how and where the main text was changed. Track changes highlight all of the modifications we made in the original paper so that they can be seen as red or green, underlined text within the manuscript, or in small bubbles on the left of the text whenever a deletion was made.
Finally, we would like to declare; that this revision has been developed in close consultation with all co-authors, and each author approved the final, revised form of this manuscript. We look forward to hearing from you regarding our re-submission, and we are open to respond to any further questions or comments you may have.
Thank you for your consideration of this manuscript!
Sincerely yours,
Ramona Dobre, MD, PhD Student
Corresponding author
RESPONSE TO REVIEWER-2 COMMENTS
We want to thank the Reviewer for the careful and thoughtful review of our manuscript and for all of the valuable suggestions he/she made. We hope that we satisfactory responded to all of the comments and made the necessary changes to the manuscript. We also want to thank the Reviewer for his/her appraisal of our review and for taking the time and energy to help us to improve the quality of our paper.
Point 1: Need to integrate meta-analysis more comprehensively in a more accurate conclusion.
Response 1: Thank you for your thoughtful observation. We integrated additional comments into the Discussion section (lines 661-666).
Point 2: Need to improve grammatical English and more thorough and comprehensive conclusion of findings.
Response 2: Lastly, but not least important, we thank you again for the suggestions that came from a meticulous read of our material. The manuscript was corrected by a colleague who is a Native English speaker. Furthermore, we checked the whole text with a Writing Assistant Program to eliminate all of the remaining grammar mistakes. Based on your observation, the Conclusion section was updated.
